# WHAT MAKES CONVOLUTIONAL MODELS GREAT ON LONG SEQUENCE MODELING?

**Yuhong Li**[1*]  **Tianle Cai**[2*]  **Yi Zhang**[3]  **Deming Chen**[1]  **Debadeepta Dey**[3]
[1]University of Illinois Urbana-Champaign, [2]Princeton University, [3]Microsoft Research.

## ABSTRACT

Convolutional models have been widely used in multiple domains. However, most existing models only use *local convolution*, making the model unable to handle *long-range dependency* efficiently. Attention overcomes this problem by aggregating *global* information based on the pair-wise attention score but also makes the computational complexity quadratic to the sequence length. Recently, Gu et al. (2021a) proposed a model called S4 inspired by the state space model. S4 can be efficiently implemented as a *global convolutional model* whose kernel size equals the input sequence length. With Fast Fourier Transform, S4 can model much longer sequences than Transformers and achieve significant gains over SoTA on several long-range tasks. Despite its empirical success, S4 is involved. It requires sophisticated parameterization and initialization schemes that combine the wisdom from several prior works. As a result, S4 is less intuitive and hard to use for researchers with limited prior knowledge. Here we aim to demystify S4 and extract basic principles that contribute to the success of S4 as a global convolutional model. We focus on the structure of the convolution kernel and identify two critical but intuitive principles enjoyed by S4 that are *sufficient* to make up an effective global convolutional model: 1) The parameterization of the convolutional kernel needs to be efficient in the sense that the number of parameters should scale sub-linearly with sequence length. 2) The kernel needs to satisfy a decaying structure that the weights for convolving with closer neighbors are larger than the more distant ones. Based on the two principles, we propose a simple yet effective convolutional model called Structured Global Convolution (`SGConv`). `SGConv` exhibits strong empirical performance over several tasks: 1) With faster speed, `SGConv` surpasses the previous SoTA on Long Range Arena and Speech Command datasets. 2) When plugging `SGConv` into standard language and vision models, it shows the potential to improve both efficiency and performance.

## 1 INTRODUCTION

Handling Long-Range Dependency (LRD) is a key challenge in long-sequence modeling tasks such as time-series forecasting, language modeling, and pixel-level image generation. Unfortunately, standard deep learning models fail to solve this problem for different reasons: Recurrent Neural Network (RNN) suffers from vanishing gradient, Transformer has complexity quadratic in the sequence length, and Convolutional Neural Network (CNN) usually only has a local receptive field in each layer.

A recently proposed benchmark called Long-Range Arena (LRA) (Tay et al., 2020b) reveals that all existing models perform poorly in modeling LRD. Notably, on one spatial-level sequence modeling task called Pathfinder-X from LRA, all models fail except a new Structured State Space sequence model (S4) (Gu et al., 2021a). The S4 model is inspired by the state space model widely used in control theory and can be computed efficiently with a special parameterization based on the Cauchy kernel. The exact implementation of the S4 model can be viewed as a *(depthwise) global convolutional model* with an involved computation global convolution kernel. Thanks to the global receptive field of the convolution kernel, S4 is able to handle tasks that require LRD, such as Pathfinder (Linsley et al., 2018; Tay et al., 2020b), where classic local CNNs fail (Linsley et al., 2018; Kim et al.,

---

*Equal contribution. Work done during the internship at Microsoft Research. Code is available.

2019). Also, the use of Fast Fourier Transform (FFT) and techniques from numerical linear algebra make the computational complexity of S4 tractable compared to the quadratic complexity of attention. Together, S4 shows the potential of global convolutional models to model LRD and advances the SoTA on LRA.

Despite its accomplishments, the delicate design of S4 makes it unfriendly even to knowledgable researchers. In particular, the empirical success of S4 relies on 1) A Diagonal Plus Low Rank (DLPR) parameterization whose efficient implementation requires several numerical linear algebra tricks, 2) An initialization scheme based on the HiPPO matrix derived in prior work (Gu et al., 2020). Therefore, aiming to reduce the complications of the model and highlight minimal principles, we raise the following questions:

*What contributes to the success of the S4 model? Can we establish a simpler model based on minimal principles to handle long-range dependency?*

To answer these questions, we focus on the design of the global convolution kernel. We extract two simple and intuitive principles that contribute to the success of the S4 kernel. The first principle is that the parameterization of the global convolution kernel should be efficient in terms of the sequence length: the number of parameters should scale slowly with the sequence length. For example, classic CNNs use a fixed kernel size. S4 also uses a fixed number of parameters to compute the convolution kernel while the number is greater than classic CNNs. Both models satisfy the first principle as the number of parameters does not scale with input length. The efficiency of parameterization is also necessary because the naive implementation of a global convolution kernel with the size of sentence length is intractable for inputs with thousands of tokens. Too many parameters will also cause overfitting, thus hurting the performance. The second principle is the decaying structure of the convolution kernel, meaning that the weights for convolving with closer neighbors are larger than the more distant ones. This structure appears ubiquitously in signal processing, with the well-known Gaussian filter as an example. The intuition is clear that closer neighbors provide a more helpful signal. S4 inherently enjoys this decaying property because of the exponential decay of the spectrum of matrix powers (See Figure 2), and we find this inductive bias improves the model performance (See Section 4.1.2).

We show that these two principles are sufficient for designing a global convolutional model that captures LRD well. To verify this, we introduce a class of global convolution kernels with a simple *multiscale* structure, as shown in Figure 1. Specifically, we compose the convolution kernel by a sequence of sub-kernels of increasing sizes, yet every sub-kernel is upsampled from the same number of parameters. This parameterization ensures that the number of parameters only scales logarithmically to the input length, which satisfies the first principle. In addition, we add a decaying weight to each scale during the combination step and fulfill the second principle. We named our methods as Structural Global Convolution kernels (SGConv). Empirically, SGConv improves S4 by more than $1\%$ and achieves SoTA results on the LRA benchmark. On Speech Command datasets, SGConv achieves comparative results in the tenclass classification task and significantly better results in the 35-class classification task upon previous SoTA. We further show that SGConv is more efficient than S4 and can be used as a general purpose module in different domains. For example, a hybrid model of classic attention and SGConv shows promising performance

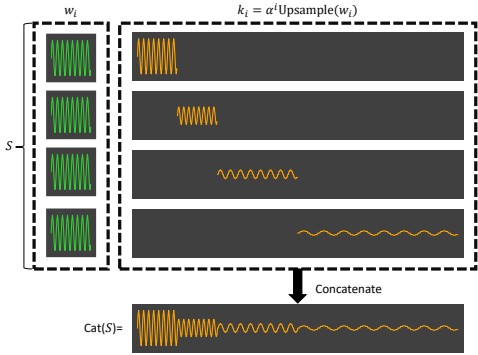

Figure 1: Illustration of the parameterization used in SGConv (Eq. (1)). The convolution kernel is composed of multi-scale sub-kernels. **Parameterization Efficiency.** Every larger sub-kernel doubles the size of the previous sub-kernel while the same number of parameters are used for every scale, ensuring a logarithmic dependency of the number of parameters to the input length. **Decaying.** We use a weighted combination of sub-kernels where the weights are decaying, and smaller weights are assigned to larger scales.

on both autoregressive language modeling and sentence classification tasks, replacing the 2D convolution kernel of the ConvNext model with 1D `SGConv` matches the performance of the original model.

## 2 RELATED WORK

**Efficient Transformers.** The Transformer architecture (Vaswani et al., 2017) has been successful across a wide range of applications (Dosovitskiy et al., 2020; Liu et al., 2021; Dong et al., 2018; Ye et al., 2022) in machine learning. However, the computation and memory complexity of Transformer scales quadratically with the input length, making it intractable for modeling long-range interactions in very long sequences. Therefore, several efficient variants of Transformer model have been proposed recently to overcome this issue (Child et al., 2019; Wang et al., 2020; Kitaev et al., 2019; Zaheer et al., 2020; Tay et al., 2020a; Peng et al., 2021; Qin et al., 2021). Nevertheless, few of these methods performed well on benchmarks such as Long Range Arena (Tay et al., 2020b), SCROLLS (Shaham et al., 2022), which require long-range modeling ability.

**(Re-)parameterization.** Parameterization is a crucial but underrated part of architecture design because different parameterizations usually provide different inductive biases. For example, weight normalization (Salimans & Kingma, 2016) parameterizes the norm and direction of the weight matrices separately and thus reaches faster convergence. On the other hand, Zagoruyko & Komodakis (2017) proposed a Dirac weight re-parameterization to train deep networks without explicit skip-connections and matched the performance of ResNets (He et al., 2016). In computer vision, several works explored using structural re-parameterization to create 2D convolution kernels. Most of these works (Ding et al., 2019; Guo et al., 2020; Ding et al., 2021; Cao et al., 2022) are limited to the vision domain and utilize only short-range convolution kernels (e.g., $7 \times 7$) except for the line of work based on 2D Fourier operators (Rao et al., 2021; Guibas et al., 2021) and the line of work based on continuous convolutional kernel (Romero et al., 2021b;a; 2022). Our `SGConv` kernel is a special parameterization of global convolution kernels that tackles LRD and showcases the extensibility of re-parameterized kernels.

**State Space Models.** The state space model (SSM) uses a set of linear differential equations to model physical systems with input, output, and state variables. It is widely used in control, neuroscience, and statistics. Recently, Gu et al. (2021b) introduced a deep SSM-based model that can outperform prior approaches on several long sequence modeling tasks with a specially structured state transition matrix. However, the expensive computation and memory requirements make it impractical. A followup work of Gu et al. (2021b) proposed a new parameterization of SSM (Gu et al., 2021a), which decomposes the state transition matrix into the sum of low-rank and normal matrices and implements SSM as a global convolutional model. Under this parameterization, the authors then combine the techniques of diagonalizing the Cauchy kernel and performing low-rank corrections with the Woodbury identity to compute the global convolution kernel. While achieving promising results, S4 is theoretically involved and practical implementations of S4 require accelerator-specific dedicated code optimization for the Cauchy kernel computation. This makes it difficult to readily implement in deep learning frameworks (Abadi et al., 2016; Chen et al., 2015; Chen, 2021; Ma et al., 2019) and hardware targets. Concurrent with this work, many state-space-based models are emerging and bringing better performance (Gu et al., 2022a; Smith et al., 2022; Hasani et al., 2022).

## 3 DESIGN OF GLOBAL CONVOLUTIONAL MODELS

We summarize the design principles that enable the global convolutional model to be both efficient and effective. Then we introduce the proposed Structured Global Convolution (`SGConv`) based on the highlighted principles.

### 3.1 DESIGN PRINCIPLES

The two intuitive design principles that contribute to the success of S4 are efficient parameterization and decaying structure.

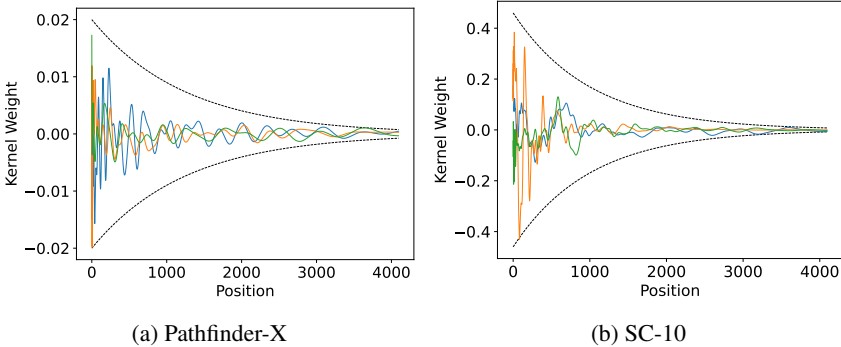

(a) Pathfinder-X                    (b) SC-10

Figure 2: Visualization of S4 kernels on (a) Pathfinder-X and (b) Speech Command 10-class. The values in the convolution kernel exhibit a decaying behavior. We only plot the first 4096 positions for better illustration.

**Efficient Parameterization.**   Different from local convolution, where the kernel size is fixed, global convolution requires a kernel size that is the same as the sentence length. Naive parameterization of convolution kernel as classic local convolutions is therefore intractable for long sequences. For instance, the Pathfinder-X task has a length of $16K$. It then impractically requires $4M$ parameters for a single layer to model the depth-wise global convolution kernel with a standard channel size of $256$. Thus, an efficient convolution kernel parameterization is necessary, especially when the sentence is extremely long. For example, S4 takes a well-designed Normal Plus Low-Rank (NPLR) parameterization to model the whole kernel with two special matrices where the number of parameters is fixed.

**Decaying Structure.**   Apart from the efficiency of the parameterization, we find that a decaying structure of the convolution kernel provides a good inductive bias to long-sequence modeling and contributes to the performance (See Section 4.1.2 for detailed ablation study). Concretely, the magnitude of the value in the convolution kernel should decay so that more weight is assigned to the close neighbors. S4 model inherently satisfies this property because the $k$-th element of the kernel of S4 is $\mathbf{CA}^k\mathbf{B}$ and the operator norm of the power of a matrix decays exponentially:

**Fact 1.** *For a square matrix $\mathbf{A}$, the operator norm $\left\|\mathbf{A}^k\right\|_2 \leq \|\mathbf{A}\|_2^k$. In particular, if $\|\mathbf{A}\|_2 < 1$, $\left\|\mathbf{A}^k\right\|_2$ decays exponential to $k$, so $\left\|\mathbf{CA}^k\mathbf{B}\right\|_2 \leq \|\mathbf{C}\|_2 \left\|\mathbf{A}^k\right\|_2 \|\mathbf{B}\|_2$ also decays exponentially.*

We can also directly observe the decaying structure of S4 in different tasks in Figure 2.

### 3.2   SGCONV

Putting the two principles altogether, we propose a simple global depth-wise convolution, dubbed Structured Global Convolution (`SGConv`), based on multiscale sub-kernels and weighted combinations. (See Figure 1). We will first introduce the parameterization of the convolutional kernel and then present how to build a global convolutional model with this kernel.

**Parameterization of SGConv Kernel.**   Formally, let $L$ be the length of the input sequence, the convolutional kernel should also has length $L$. We define the parameter set of a single channel as $S = \left\{\mathbf{w}_i | 0 \leq i < \left\lceil \log_2\left(\frac{L}{d}\right)\right\rceil + 1\right\}$ where $\mathbf{w}_i \in \mathbb{R}^d$ is the parameter for $i$-th sub-kernel $k_i$, and $d$ is the dimension of the parameter. Denote the number of scales $N = \left\lceil\log_2\left(\frac{L}{d}\right)\right\rceil + 1$. We use the upsample operation, implemented as linear interpolation, to form sub-kernels of different scales. We use $\text{Upsample}_l(\mathbf{x})$ to denote upsampling $\mathbf{x}$ to length $l$ (We use `F.interpolate` function in Pytorch and set the mode to be `linear` in our implementation). We also introduce a normalization constant $Z$ to ensure the convolution operation will not change the scale of the input and a coefficient $\alpha$ to control the decaying speed. Now, we are ready to introduce the weighted combination scheme by concatenating a set of weighted sub-kernels $k_i$:

$$\text{Cat}(S) = \frac{1}{Z}\left[k_0, k_1, \cdots, k_{N-1}\right], \text{ where } k_i = \alpha^i \text{Upsample}_{2^{\max[i-1,0]}d}\left(\mathbf{w}_i\right). \qquad (1)$$

| Model | ListOps | Text | Retrieval | Image | Pathfinder | Path-X | Avg. |
|---|---|---|---|---|---|---|---|
| Transformer | 36.37 | 64.27 | 57.46 | 42.44 | 71.40 | ✗ | 54.39 |
| Sparse Trans. | 17.07 | 63.58 | 59.59 | 44.24 | 71.71 | ✗ | 51.24 |
| Linformer | 35.70 | 53.94 | 52.27 | 38.56 | 76.34 | ✗ | 51.36 |
| Reformer | 37.27 | 56.10 | 53.40 | 38.07 | 68.50 | ✗ | 50.67 |
| BigBird | 36.05 | 64.02 | 59.29 | 40.83 | 74.87 | ✗ | 55.01 |
| S4 (original) | 58.35 | 76.02 | 87.09 | 87.26 | 86.05 | 88.10 | 80.48 |
| S4 (Gu et al., 2022b) | 59.60 | 86.82 | **90.90** | **88.65** | 94.20 | 96.35 | 86.09 |
| SGConv | **61.45** | **89.20** | **91.11** | 87.97 | **95.46** | **97.83** | **87.17** |

Table 1: The performance of `SGConv` compared to other baselines on the LRA dataset. `SGConv` achieves significant improvement compared to previous methods with a more straightforward structure and faster speed (See Table 2)

It is easy to check that $\mathrm{Cat}(S)$ gives the convolution kernel with length $\sum_{i=0}^{N} 2^{\max[i-1,0]}d = 2^{N-1}d \geq L$ (See Figure 1 for an illustration), which can be truncated to $L$ if it is overlength. And the number of parameters is $Nd = O(\log L)$. The decay coefficient $\alpha$, usually chosen to be $1/2$, induces the decaying structure.

**Incorporate SGConv to Modern Architectures.** In the implementation, we compute the depthwise convolution kernel and use Fast Fourier Transform to compute the convolution in $O(L \log L)$ time (See Figure 8 for detailed illustration). We compute the normalization constant $Z$ such that the norm of the kernel is one at initialization and fix it during training. Please refer to Appendix B.2 for a Python-style pseudo-code. We can plug `SGConv` into modern architectures as a replacement of attention in Transformer or local convolution in ConvNets (See Figure 6, 7 for two examples). Due to the relaxation of the structure of the convolutional kernel, `SGConv` does not have the RNN-style reformulation as S4. Yet, `SGConv` is naturally capable of performing autoregressive generation, such as language modeling, similarly to classic causal convolutional models (Van den Oord et al., 2016; Oord et al., 2016) and Transformers. Concretely, the convolution kernel is unidirectional, where the computation at the embedding of $i$-th is only computed based on tokens before $i$, and left zero padding is used for ignoring the overlength kernel. During generation, hidden states of past tokens are cached for fast calculation of the next token with a single convolution step. Due to the simplicity of the parameterization, `SGConv` kernel is easy to compute and more efficient than the S4 kernel, as shown in Section 4.1.3.

## 4 EXPERIMENTS

In this section, we first test the effectiveness of `SGConv` on two standard long sequence modeling tasks, i.e., Long Range Arena (Tay et al., 2020b) and Speech Commands (Warden, 2018), and compare it with S4 and other baselines. We also conduct ablation studies over the decay speed and scale dimension $d$ and evaluate the speed of `SGConv` on LRA. Further, we explore the possibility of plugging the global convolutional layer into standard models as a *general-purpose component* for capturing long-range dependency. For language tasks, we find that replacing half of layers of Transformer with a certain strategy with `SGConv` block will not hurt performance, while the complexity of those layers improves from $O(L^2)$ to $O(L \log L)$. On ImageNet, we replace the $7 \times 7$ convolution in ConvNext (Liu et al., 2022) with `SGConv` and show comparative or better performance.

### 4.1 LONG RANGE ARENA

Long Range Arena benchmark (Tay et al., 2020b) is a suite of six tasks consisting of sequences ranging from 1K to 16K tokens, encompassing a wide range of data types and modalities such as text, natural, synthetic images, and mathematical expressions requiring similarity, structural, and visual-spatial reasoning.

### 4.1.1 RESULTS

We show the experimental results in Table 1 with several baseline methods (Vaswani et al., 2017; Child et al., 2019; Wang et al., 2020; Kitaev et al., 2019; Zaheer et al., 2020; Gu et al., 2021a; 2022b).

| | Sequence length | 256 | 512 | 1024 | 2048 | 4096 | 8192 | 16384 |
|---|---|---|---|---|---|---|---|---|
| Inf. CPU | S4 | 29.4 | 81.7 | 158.3 | 306.9 | 594 | 1156.9 | 2274.0 |
| | SGConv | **23.8** | **56.2** | **108.7** | **211.3** | **409.3** | **789.5** | **1559.3** |
| Inf. GPU | S4 (w/o opt) | 2.7 | 2.7 | 4.4 | 7.9 | 15.2 | 32.7 | 64.5 |
| | S4 (w. opt.) | 1.6 | 1.9 | 3.1 | 5.4 | 10.0 | 22.3 | 44.3 |
| | SGConv | **1.2** | **1.3** | **2.3** | **4.4** | **8.5** | **19.8** | **39.4** |
| BP GPU | S4 (w/o opt) | 4.1 | 5.7 | 10.2 | 19.4 | 38.1 | 80.1 | 161.2 |
| | S4 (w. opt.) | 3.5 | 4 | 6.6 | 11.9 | 22.6 | 48.9 | 97.8 |
| | SGConv | **2.0** | **2.7** | **5.0** | **9.6** | **18.6** | **41.2** | **82.5** |

Table 2: Comparison of the inference and backpropagation time (ms/batch) of S4 and `SGConv` blocks (number of channels 128, batch size 64) on CPU and GPU. Note that the parameterization in S4 requires a customized CUDA kernel to improve the efficiency (refer to opt. in the Table). Nevertheless, `SGConv` still *always* surpasses S4 even compared to the optimized CUDA kernel.

`SGConv` achieves a 1% improvement in average accuracy upon well-tuned S4 variants introduced in Gu et al. (2022b). Notably, `SGConv` is guided by the two intuitive principles and has a much simpler structure than S4 (Gu et al., 2022b). The detailed implementation settings can be found in Appendix A.1.

### 4.1.2 ABLATION STUDY ON IMDB

We conduct ablation studies on the IMDB byte-level document classification task in the LRA benchmark. We mainly focus on two aspects: 1) The speed of decaying and 2) The parameter dimension $d$ of each scale. For simplicity, in the standard `SGConv` formulation (Eq. (1)), we fix the decay coefficient $\alpha = 1/2$ and only tune the dimension $d$. However, the actual decay speed as a function of the position in the kernel depends both on $\alpha$ and $d$, making it hard to conduct ablation studies. Thus, we use a slightly different convolution kernel that disentangles the decay speed and the dimension of each scale:

$$\text{Cat*}(S) = \frac{1}{Z}\left[k_0, k_1, \cdots, k_{N-1}\right] \odot \left[\frac{1}{1^t}, \frac{1}{2^t}, \cdots, \frac{1}{L^t}\right], \text{ where } k_i = \text{Upsample}_{2^{\max[i-1,0]}d}\left(\mathbf{w}_i\right).$$

(2)

$t$ here then controls the decay speed, which is independent of each scale's dimension. We conduct two sets of experiments: 1) Fix $d = 8$, vary $t$ from 0 (which means no decay) to 2, and 2) Fix $t = 1$, vary $d$ from 1 to 64. Figure 3 reports the accuracies in different settings. We can observe that 1) The decay structure is crucial for getting good performance, and 2) In a reasonable range, $d$ has less impact on the performance than $t$. Nevertheless, we observe a trend of performance drop when increasing $d$ from 8 to 64. Experiments on larger $d$ show worse performance, which can be attributed to overfitting.

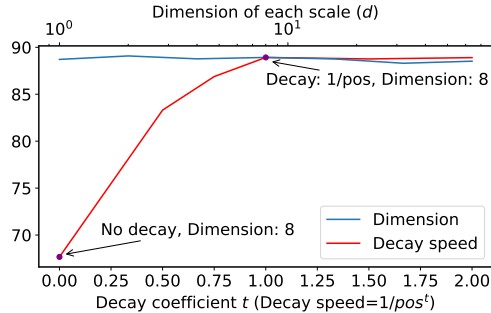

Figure 3: Ablation study on the effect of decay speed and hidden dimension of each scale on IMDB dataset. $pos \in [1, L]$ refers to the position in the convolution kernel. We observe: 1) The decay structure is crucial for getting good performance; 2) In a reasonable range, $d$ (Dimension) has less impact on the performance than $t$ ($t \in [0, 2.0]$).

### 4.1.3 SPEED COMPARISON

In Table 2, we compare the computation speed of the S4 kernel and `SGConv` kernel in different settings. Due to its simplicity, `SGConv` is faster than S4 for any sentence length. `SGConv` is about 50% faster than the vanilla implementation of the S4 kernel and is 15% faster than the optimized CUDA kernel implementation without resorting to optimized CUDA kernels.

## 4.2 SPEECH COMMANDS

The Speech Command (SC) dataset (Warden, 2018) is a 35-class dataset of 1 second (16000 HZ sampling rate) spoken words in English. However, followup works (Kidger et al., 2020; Gu et al., 2021b; Romero et al., 2021b;a) adopted a smaller 10-class subset of SC. And works (Romero et al., 2021a; Gu et al., 2021b) on the SC dataset specifically use pre-processing such as MFCC features. Our baselines are obtained from (Gu et al., 2021a; 2022a). Note that besides SSM-based models, there is no strong baseline for raw waveform classification using either the 10-class or the full dataset. And SSM-based methods also show the ability to perform 0-shot testing at lower sampling rate such as 8000 Hz. Table 3 shows that the `SGConv` yields better results compared to the SSM-based method among 4 out of 5 tasks. Notably, for the original SC (35-class), `SGConv` achieves marginally higher accuracy for raw-sequence classification and significantly better results (+2.40%) compared to the existing SoTA method.

| 10-cls | Transformer | Performer | NRDE | CKConv | WaveGAN-D | S4 | S4* | SGConv |
|---|---|---|---|---|---|---|---|---|
| MFCC | 90.75 | 80.85 | 89.8 | **95.3** | ✗ | 93.96 | 92.05 | 94.91 |
| 16000HZ | ✗ | 30.77 | 16.49 | 11.6 | 71.66 | 98.32 | **97.98** | 97.52 |
| 8000HZ (0-shot) | ✗ | 30.68 | 15.12 | 65.96 | ✗ | 96.30 | 91.83 | **96.03** |

| 35-cls | InceptionNet | ResNet-18 | XResNet-50 | ConvNet | S4D | S4 | S4* | SGConv |
|---|---|---|---|---|---|---|---|---|
| 16000HZ | 61.24 | 77.86 | 83.01 | 95.51 | 96.25 | 96.08 | 96.27 | **96.42** |
| 8000HZ (0-shot) | 5.18 | 8.74 | 7.72 | 7.26 | 91.58 | 91.32 | 91.89 | **94.29** |

Table 3: Speech Command classification results compared to existing methods. * We carefully reproduce the S4 method based on the released code[1]. Since the latest version removed 10-class experiments settings, we utilized a earlier version[2]. The results suggest that for the SC 35-classification, `SGConv` achieves SoTA on both full length task and 2X sampling rate, zero-shot task.

## 4.3 FURTHER APPLICATIONS OF SGCONV

We further study `SGConv` as a generic network architecture *drop-in* component targeting tasks in language modeling and computer vision. In Section 4.3.1 we present an efficient mixture of attention and `SGConv` layers architecture that replaces half of the attention blocks in the Transformer with the `SGConv` blocks. We demonstrate the potential of utilizing such a model for long text processing. In Section 4.3.2, we incorporate `SGConv` (1D) into ConvNeXt (Liu et al., 2022). Surprisingly, `SGConv` achieves comparable or even better results compared to several SoTA CNN and Vision Transformer models by treating the 2D features as a 1D sequence.

### 4.3.1 LANGUAGE TASKS

**Language modeling.** We propose the `SGConv` block (shown in Figure 6) which is similar to the Attention block in Transformer (Vaswani et al., 2017). `SGConv` block enjoys both $O(L\log(L))$ time complexity and space complexity. We benchmark the inference time and GPU memory usage of both `SGConv` and Attention in Table 7. When the sequence length is 1024, `SGConv` block is ~2.1X faster than the Attention block. For language modeling, we utilize the feature of `SGConv` to directly process the long sequences. The

| Model | Valid. | Test |
|---|---|---|
| LSTM+Hebb. | 29.0 | 29.2 |
| 16L Transformer-XL | - | 24.0 |
| 16L SGConv+SAttn | 21.90 | **22.83** |
| Adaptive Input | - | 18.7 |
| S4 | - | 20.95 |
| 18L Transformer-XL | - | **18.3** |
| 18L Transformer-XL* | 18.16 | 18.75 |
| 18L SGConv+SAttn | 18.10 | 18.70 |

Table 4: Performance comparison on WikiText-103.

---

[1] `https://github.com/HazyResearch/state-spaces`
[2] `https://github.com/HazyResearch/state-spaces/tree/` `307f11bba801d5734235a1791df1859f6ae0e367`

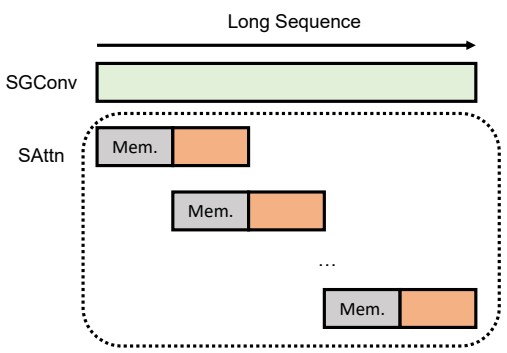

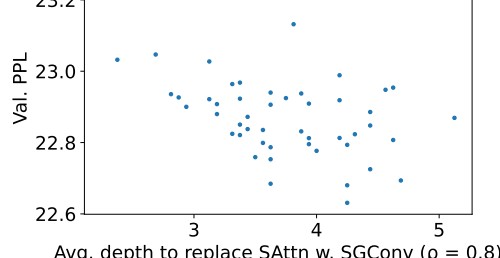

(a) Illustration of `SGConv` and Transformer-XL style Short Attention used in language modeling task. `SGConv` directly processes the full length sequence.

(b) The depth to replace SAttention with `SGConv` vs. validation PPL on WikiText-103

Figure 4: Incorporating `SGConv` to Transformer models in language tasks.

|  | MNLI-m/mm | QNLI | QQP | SST | CoLA | STS | Avg. |
|---|---|---|---|---|---|---|---|
| BERT | **84.93/84.91** | **91.34** | 91.04 | **92.88** | 55.19 | 88.29 | 84.08 |
| SGConvBERT | 84.78/84.70 | 91.25 | **91.18** | 92.55 | **57.92** | **88.42** | **84.40** |

Table 5: Performance comparison of BERT and `SGConvBERT` on GLUE dataset. `SGConvBERT` is comparable with BERT while being more efficient. We exclude MRPC and RTE datasets in GLUE because their sizes are too small ($< 5K$ training samples).

Attention block only targets the short range data termed SAttention. We illustrate the structure in Figure 4a. Furthermore, we investigate the strategy to replace the Attention blocks with `SGConv` blocks. We generate 50 architectures with 8 `SGConv` blocks and 8 Attention blocks where the order is shuffled. We denote the average depth to replace the Attention blocks as: $\sum_{i=0}^{N_{SGConv}} \mathrm{idx}_i / N_{total}$ where the idx denotes the $i$th `SGConv` depth position. $N_{SGConv} = 8$ and $N_{total} = 16$ in this case. The results in Figure 4b suggest that when fixing the number of `SGConv` layer, models achieve better performance by placing `SGConv` blocks in *deeper* layers. Guided by the strategy, we handcraft two Transformer-XL (Dai et al., 2019) style models. (1) 16-layer: {A, A, A, C}×2 + {A, C, C, C}×2. (2) 18-layer: {A, A, C}×3 + {A, C, C}×3. A denotes SAttention and C denotes `SGConv`. ×$N$ denotes repeating the order of layers for $N$ times. We test the model on WikiText-103 (Merity et al., 2016) which is a wide-used language modeling benchmark with an average length of 3.6K tokens per article. We set both the attention and memory length to 384 for 18L model and 192 for 16L model. The length of input sequence is 3092 which can be processed by `SGConv` directly. We show the results in Table 4. Our results suggest that when the attention range is short, the 16L model outperform the baseline with -1.17 perplexity. For the 18L model, our model achieves 18.70 perplexity. Note that we use a smaller and affordable batch size (16) for training. Under the same setting, our model gains slightly better perplexity than Transformer-XL (-0.05). Our results show the potential of adopting `SGConv` as part of the language model for long range language sequence processing.

**Sentence classification.** We combine the `SGConv` block with the BERT model (Devlin et al., 2018). Concretely, we utilize the 12-layer {A, A, C}×2+{A, C, C}×2 model. The pretraining is conducted on BooksCorpus (Zhu et al., 2015) and English Wikipedia (Foundation). We then fine-tune the model on the GLUE benchmark (Wang et al., 2019). To avoid the instability of fine-tuning on small datasets, we only test on tasks with more than $5K$ training samples. We follow the training and fine-tuning pipeline of Ke et al. (2020) (BERT-A in Table 1 of Ke et al. (2020)) and report the average accuracy of 5 different random seeds. `SGConvBERT` achieves comparable performance to the original BERT model, while the `SGConv` layer is more efficient than the attention layer.

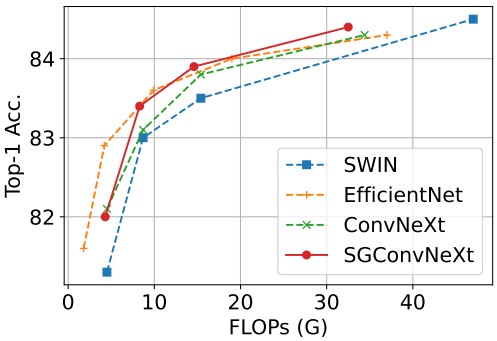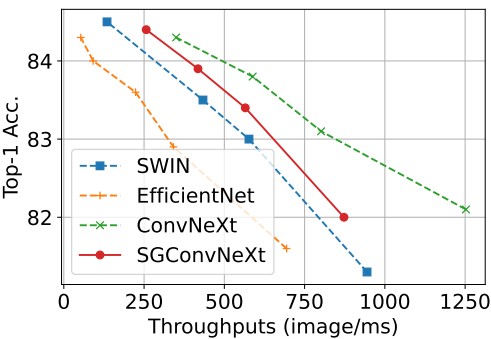

Figure 5: Comparison of ImageNet-1k Top-1 accuracy with SoTA works. Left: Top-1 Accuracy vs. FLOPs. Right: Top-1 Accuracy vs. Throughputs.

### 4.3.2 IMAGE CLASSIFICATION

We also evaluate the adaptability of `SGConv` by applying it on large-scale image classification. We conduct experiments on ImageNet-1k (Deng et al., 2009) which consists of more than 1.28 million high-resolution training and 50,000 validation images. We replace the $7 \times 7$ 2D convolutional kernels with `SGConvs` in ConvNeXt (Liu et al., 2022) denoted as `SGConvNeXt`. The block designs of `SGConvNeXt` are shown in Figure 7. Note we train the SGConveNeXt-Tiny/Small/Base/Large using hyperparameter settings from ConvNeXt[4] without any changes. By treating the 2D features as sequences, our `SGConvNeXt` achieves better results compared to existing SoTA methods such as EfficientNets (Tan & Le, 2019), Swin Transformers (Liu et al., 2021) (shown in Figure 5). Note that Vision Transformer (Dosovitskiy et al., 2020) and its variants (Touvron et al., 2021a;b; Yu et al., 2022) adopt patching techniques that can lead to a quadratic increase in complexity with image size. Also, patching is incompatible with dynamic input resolutions while `SGConvNeXt` processes the data globally. We list several interesting directions that can be explored for future work: 1) Optimization for the long-range convolution: we noticed that though FFT theoretically requires less FLOPs than plain convolution, the throughput drops empirically. One reason is that there is no optimized CUDA implementation for 1D long-range convolution and can be a good direction for future work. 2) Optimized hyperparameters and data augmentation methods: ConvNeXts' hyperparameters are tuned for maximum performance, which may not be ideal for `SGConvNeXt`. 3) `SGConv` for vision reasoning tasks: we show that `SGConv` is powerful for long-range synthetic reasoning tasks and large-scale classification tasks. It could be effective in visual reasoning applications such as Vision-Language Reasoning (Johnson et al., 2017; Zhu et al., 2020) with great potential.

## 5 DISCUSSION

In this paper, we attempt to answer the question of what makes convolutional models great again on long sequence modeling and summarize two principles contributing to the success. Based on the principles, we propose a simple and intuitive global convolutional model `SGConv` that has both direct implications and solid performance. Concurrent to our work there are also attempts to simplify the S4 model by restricting the state transition matrix to be diagonal (Gu et al., 2022a; Gupta, 2022). The proposal by Gu et al. (2022a) incorporates an intricate approach to parameterization and initialization schemes compared to our paper. Their method provides insights into the S4 phenomenon from a state-space-model perspective. Instead, we hope our simpler principles and non-SSM-based model can open up a direction for general audiences to understand and try global convolution as a general-purpose module for tackling long-range dependency. This potential has been shown in a very recent paper (Ma et al., 2022) concurrent to our work, where the authors incorporate an exponential moving average layer to a Transformer-like model and achieve promising performance over several long sequence modeling tasks. The exponential moving average layer is a particular type of global convolution layer that naturally satisfies our two principles. We believe that similar global convolutional modules will emerge in the future as long-range dependency becomes increasingly critical for sequence modeling.

ACKNOWLEDGEMENTS

We extend our gratitude to the anonymous reviewers for dedicating their time and expertise to provide constructive feedback and suggestions, which significantly enhanced the quality of this paper. We also express our appreciation to the Program Chairs and Area Chairs for their careful review and valuable comments. Special thanks go to Sébastien Bubeck, Arturs Backurs, Gustavo de Rosa, Di He, and Cong 'Callie' Hao for their valuable suggestions and support.

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

# A   DETAILED EXPERIMENTAL RESULTS

## A.1   LONG RANGE ARENA

Here we report the detailed implementation of the LRA experiments. We use the concatenation style combination of sub-kernels in all experiments and mildly tune the dimension of each scale. Since the `SGConv` exhibits a strong ability to fit data, we slightly increase the dropout for some tasks to prevent overfitting. Table 6 lists the detailed hyperparameters used in LRA. In most experiments, we set $\alpha$ to $1/2$, which approximately decays in speed $1/pos$. Experiments on flattened 2D images require some special modification of the kernel. We hypothesize that it is because images require more subtle inductive bias. For the experiment on the Image dataset, we use the disentangled version of parameterization and combination weights as described in Section 4.1.2 and set the decay speed to be $1/pos$. For the experiment on the Pathfinder-X task, we initialize convolution kernels in different channels with cosine waves with different frequencies and randomly assign $\alpha$ ranging from 1 to $1/3$ to different channels. Both these modifications bring about $1\%$ improvement compared to standard fixed $\alpha = 1/2$ and random initialization. The remaining hyperparameters and experimental settings are same to Gu et al. (2022a) which can be found in the Github repo[1].

|            | ListOps | Text  | Retrieval | Image | Pathfinder | Path-X |
| ---------- | ------- | ----- | --------- | ----- | ---------- | ------ |
| Acc.       | 61.45   | 89.20 | 91.11     | 87.97 | 95.46      | 97.83  |
| Scale dim. | 1       | 2     | 1         | 32    | 32         | 64     |
| Dropout    | 0       | 0     | 0         | 0.2   | 0.2        | 0      |

Table 6: Hyperparameters used in LRA experiments.

## A.2   SPEECH COMMAND

For Speech Command 10-class task, we use the same training setting from Gu et al. (2021a) earlier version Github repo[2]. For Speech Command 35-class task, we use the training setting from the Github repo[1]. The scale dimension of `SGConv` is 32.

## A.3   LANGUAGE TASK

Our implementation for Language Task is based on the project [3]. For the 16-L model, we utilize 3072 as the sequence length for `SGCONV` and 192 as both the attention and memory length for SAttention. For the 18-L model, we utilize 3072 as the sequence length for `SGCONV` and 384 as both the attention and memory length for SAttention. The `SGConv` has 96 as the scale dimension. We adopt the training settings from the above mentioned project 3 except the batch size which is reduced to 64. The `SGConv` block is shown in Figure 4.

---

[3]`https://github.com/NVIDIA/DeepLearningExamples/tree/master/PyTorch/`
`LanguageModeling/Transformer-XL`

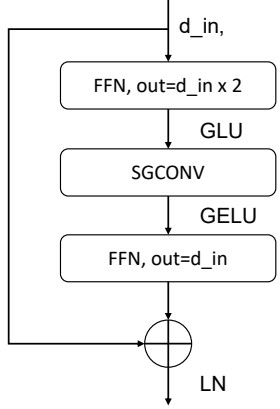

Figure 6: SGConv block

|  |  | 256 | 512 | 1024 | 2048 | 3072 |
|---|---|---|---|---|---|---|
| Attn. Block | Inf. (ms/batch) | **2.6** | 7.3 | 23.2 | 91.7 | ✗ |
|  | Mem. (GB) | **2.6** | 3.9 | 7.9 | 23.9 | OOM |
| SGConv Block | Inf. (ms/batch) | 2.7 | **5.4** | **10.9** | **21.8** | **43.6** |
|  | Mem. (GB) | **2.6** | **3.4** | **5.2** | **8.7** | **15.7** |

Table 7: Comparison of inference time and GPU memory utilization with Attention blocks. `SGConv` has significantly less memory usage and faster inference speed when the sequence increases.

## A.4 IMAGE CLASSIFICATION

We use the training settings in the work Liu et al. (2022)[4]. Since the `SGConvNeXt` has several downsampling layers, we fixed the scale to 5 and the scale dimensions are calculated based on the flattened features length of the corresponding layers. The structure is shown in Figure 7. The results are shown in Table 8. The visualization of the `SGConvNeXt`-Base outputs are shown in Figure 9. The visualization of the `SGConv` kernels at different stages are shown in Figure 10.

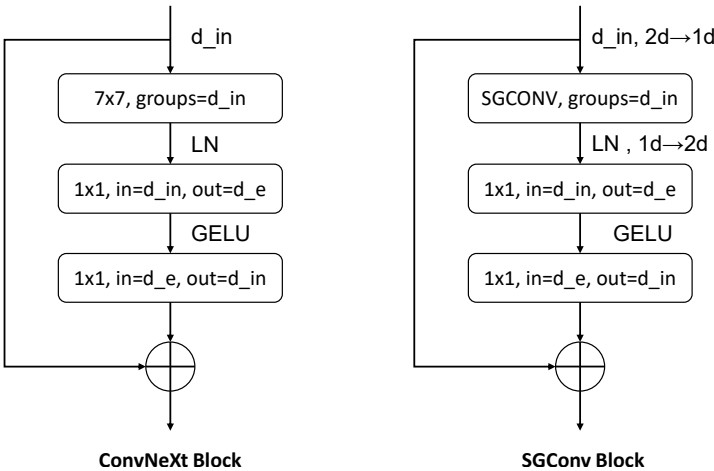

Figure 7: SGConvnext

---

[4]https://github.com/facebookresearch/ConvNeXt

| model | FLOPs | throughput (image/s) | params | Acc. |
|---|---|---|---|---|
| Swin-T | 4.5G | 944.5 | 29M | 81.3 |
| Swin-S | 8.7G | 576.8 | 50M | 83.0 |
| Swin-B | 15.4G | 433.4 | 88M | 83.5 |
| Swin-B$_{384^2}$ | 47.0G | 134.6 | 88M | 84.5 |
| ConvNeXt-T | 4.5G | 1252.6 | 29M | 82.1 |
| ConvNeXt-S | 8.7G | 801.4 | 50M | 83.1 |
| ConvNeXt-B | 15.4G | 588.3 | 89M | 83.8 |
| ConvNeXt-L | 34.4G | 349.8 | 198M | 84.3 |

| model | FLOPs | throughput (image/s) | params | Acc. |
|---|---|---|---|---|
| EffNet-B3$_{300^2}$ | 1.8G | 693.9 | 12M | 81.6 |
| EffNet-B4$_{380^2}$ | 4.2G | 341.5 | 19M | 82.9 |
| EffNet-B5$_{456^2}$ | 9.9G | 223.5 | 30M | 83.6 |
| EffNet-B6$_{528^2}$ | 19.0G | 91.5 | 43M | 84.0 |
| EffNet-B7$_{600^2}$ | 37.0G | 52.9 | 66M | 84.3 |
| SGConvNeXt-T | 4.3G | 872.6 | 29M | 82.0 |
| SGConvNeXt-S | 8.3G | 565.3 | 51M | 83.4 |
| SGConvNeXt-B | 14.6G | 417.9 | 90M | 83.9 |
| SGConvNeXt-L | 32.5G | 256.7 | 200M | 84.4 |

Table 8: Comparison of ImageNet-1k Top-1 accuracy with SoTA works.

# B DETAILED IMPLEMENTATION

## B.1 ILLUSTRATION OF SGCONV MODULE

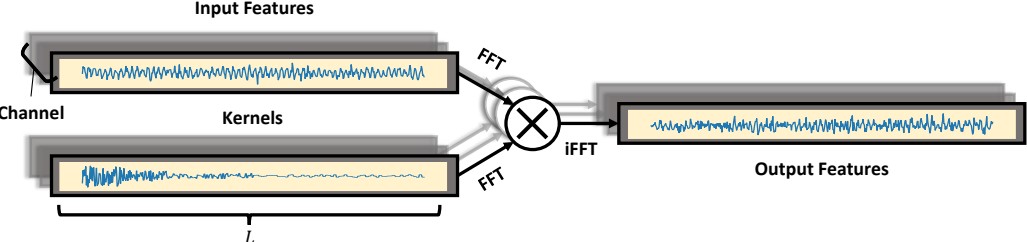

Figure 8: Implementing SGConv with FFT. We first compute the convolutional kernels for each channel as described in Section 3.2, and apply the depth-wise global convolution to the input features.

## B.2 PYTHON STYLE PSEUDO-CODE

```python
# Parameters
kernel_param_list = [] # w_i
for _ in range(num_scales):
    kernel_param_list.append(
        nn.Parameter(torch.randn(hidden_dim, kernel_dim))
    ) # size: h * d

# Compute global convolution kernel
kernel_list = [] # k_i
for i in range(num_scales):
    kernel = F.interpolate(
        kernel_param_list[i],
        scale_factor = 2**max(0, i-1),
        mode = "linear"
    ) * 0.5 ** i # alpha = 0.5
    kernel_list.append(kernel)
# The computed kernel, size: h * (d * 2^{s-1})
k = torch.cat(kernel_list, dim=-1)

# Normalize kernel
if is_init: # Compute the norm at initialization
    kernel_norm = k.norm(dim=-1, keepdim=True).detach()
k = k / kernel_norm
```

```python
# Use kernel to compute global convolution
# x: batch_size * hidden_dim * seq_len
L = x.size(-1)
# Truncate kernel if it is too long
k = k[..., :L]
# Use FFT to compute convolution
x_f = torch.fft.rfft(x, n=2*L)
k_f = torch.fft.rfft(k, n=2*L)
y_f = torch.einsum("b h l, h l -> b h l", x_f, k_f)

# Inverse FFT to get the result
y = torch.fft.irfft(y_f, n=2*L)[..., :L]
```

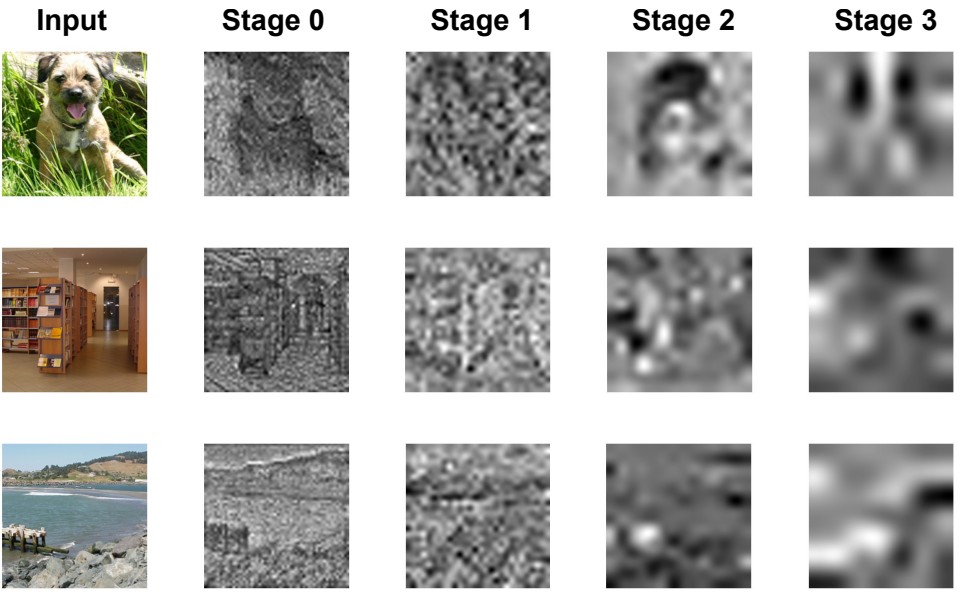

Figure 9: Visualization of the intermediate features of `SGConvNeXt` on ImageNet-1k dataset.

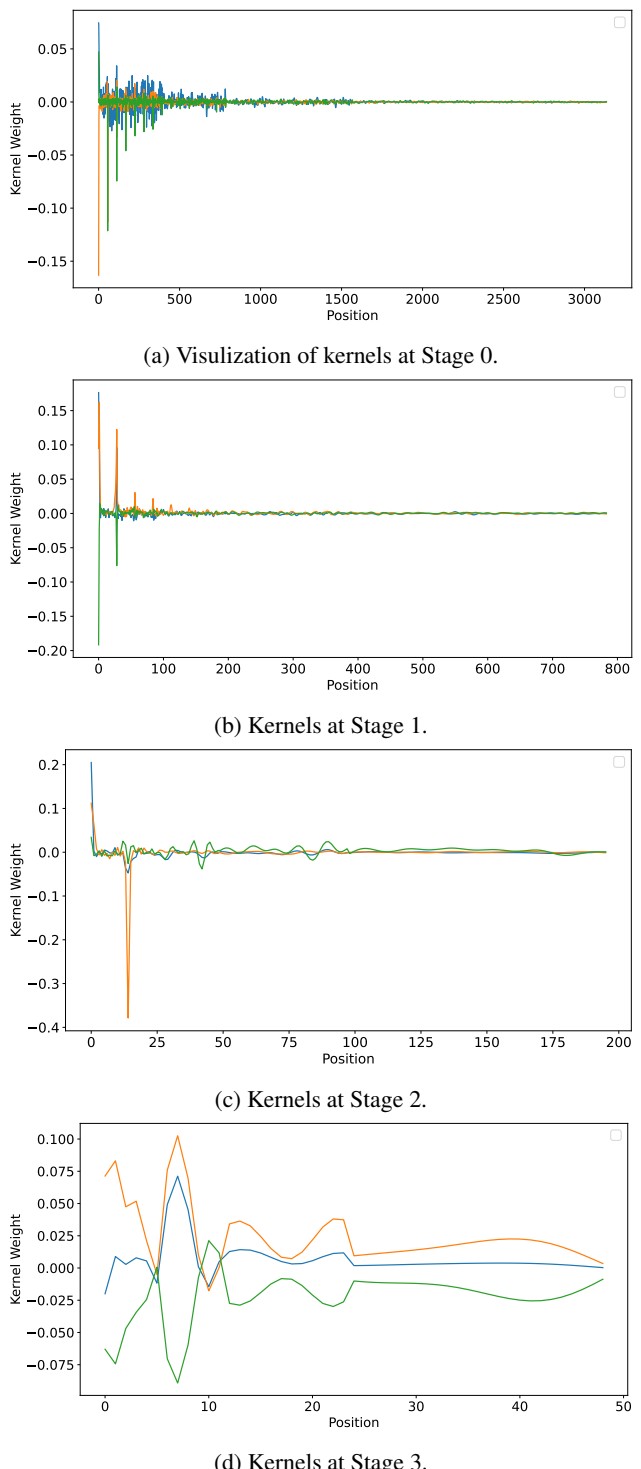

(a) Visulization of kernels at Stage 0.

(b) Kernels at Stage 1.

(c) Kernels at Stage 2.

(d) Kernels at Stage 3.

Figure 10: Kernels in `SGConvNeXt` at different stages.

## C  NEURAL ARCHITECTURE SEARCH PERSPECTIVE OF SGCONV

Neural architecture search (NAS) is an automated process for discovering a neural network's optimal architecture or structure for a particular task. NAS typically involves searching through a large

space of possible network architectures using combination algorithms, such as reinforcement learning (Zoph et al., 2018), evolutionary algorithms (Real et al., 2019), or Bayesian optimization (Kandasamy et al., 2018). In recent years, there has been a proliferation of research aimed at designing traditional convolutional neural networks with local convolution (Li et al., 2021; Lin et al., 2021; Li et al., 2023). These works primarily focus on optimizing the networks' structures to improve their performance. From the perspective of NAS, the `SGConv` can be interpreted as a kernel-level fine-grained search for the distribution of parameters by utilizing parameterization. Furthermore, the `SGConv` has shown that the global convolution kernel exhibits sparsity and can be pruned (Fig. 10), meaning that the effective kernel length can be automatically determined through the training phase. These findings can potentially spark further research and development in the field. Another simple approach we explore in NAS is the combination of Attention and `SGConv` through a mixture model (Section 4.3.1). This approach is both intuitive and efficient and has the potential to improve the performance of neural network architectures further.

