# OpenReview forum: "What Makes Convolutional Models Great on Long Sequence Modeling?"
_ICLR.cc/2023/Conference — ICLR 2023 poster_

### Official Review · Reviewer_sRmA · 2022-10-24

**Confidence:** 4
**Correctness:** 4
**Technical Novelty And Significance:** 3
**Empirical Novelty And Significance:** 4
**Recommendation:** 8

**Clarity, Quality, Novelty And Reproducibility:**

Clarity: very clear
Novely: sufficiently novel
Reproducibility: should be reproducible

**Strength And Weaknesses:**

Pros:
* A simplified model that improves over the recently proposed state-of-the-art model for long range dependency modeling
* Proposed model is more accurate and computationally more efficient.  Establishes a new state of the art on the Long Range Arena benchmark

Cons:
N/A


**Summary Of The Paper:**

The paper follows on the success of the recently proposed structured state space based sequence model (S4) for long range dependency modeling, and proposes a simplified model that keeps two key characteristics of the S4 model, namely its parameter efficiency when modeling long sequences, and a decaying structure of weights where weights on nearby context are larger than weights for far out context.  Due to its simplicity the proposed model, termed SGConv, is more efficient to compute as compared to S4 while achieving better accuracy on long range dependency modeling tasks of the long range arena (LRA) benchmark.

**Summary Of The Review:**

see above

---

> ### Author Response · Authors · 2022-11-14
> **Response to Reviewer sRmA**
>
> We appreciate that the reviewer enjoys our paper, and we have made several updates to improve it further, as mentioned in the general response.

---

### Official Review · Reviewer_gGtF · 2022-10-25

**Confidence:** 4
**Correctness:** 4
**Technical Novelty And Significance:** 3
**Empirical Novelty And Significance:** 3
**Recommendation:** 6

**Clarity, Quality, Novelty And Reproducibility:**

The paper is easy to follow, and the proposed approach is simple and effective which provides certain novelty.

**Strength And Weaknesses:**

Strength: the proposed approach is simple and effective for long-form tasks.
Weakness: the main benefits are demonstrated on the long-range arena dataset, but the benefit on LM and image classification tasks are a bit limited.

**Summary Of The Paper:**

This work proposes a convolution module capable of scaling up to long sequences. In particular the approach uses a convolution layer and upsample the kernels to process longer sequences. The approach also applies a larger weight on kernels with fewer upsampling to emphasize local information. The proposed approach shows good results on the long-range arena dataset. Further ablation studies exhibit the importance of weighting kernels that correspond to local information.

**Summary Of The Review:**

This work proposed a simple and effective way of using convolution modules on long sequences, and demonstrated good results on the long-range arena dataset. The overall quality of the work can be further improved by providing more empirical evidence of the approach’s benefit.

---

> ### Author Response · Authors · 2022-11-14
> **Response to Reviewer gGtF**
>
> We appreciate the reviewer’s helpful comments. Besides Long Range Arena, SGConv also achieves SoTA on Speech Command 35-classification task SoTA on both full-length task and 2X sampling rate, zero-shot task. Our main aim here was to really study and simplify convolutional operations for long-range tasks. In future work, we will keep improving SGConv for Language Modeling and Image Classification tasks, especially in combination with architecture search.

---

### Official Review · Reviewer_WDs3 · 2022-10-28

**Confidence:** 3
**Correctness:** 3
**Technical Novelty And Significance:** 3
**Empirical Novelty And Significance:** 2
**Recommendation:** 8

**Clarity, Quality, Novelty And Reproducibility:**

Clarity: not very clear at describing details of SGConv
Reproducibility: easy to produce

**Strength And Weaknesses:**

Strength
1. Simplification. The author simplifies the design complexity of the global convolution layer based on two principles defined in S4. Potentially, this could make the study in this work more accessible to the community dealing with long context modeling tasks.
2. Experiments are intensive, covering the commonly used LRA data sets, speech, vision and language modeling tasks and comparing with baseline, Transformer/BERT based models.

Weaknesses
1. Section 3.2 could be clearer in describing the details of SGConv when dealing with sequential input like how the convolution actually operate given input sequence and weight w, and how the operation deals with variable length inputs.
2. (more alike a suggestion) The results on speech commands data set is not that convincing because the samples are all 1 second audios. It would be recommended if the authors can do experiments on Automatic Speech Recognition or Speech translation tasks to show the effectiveness on audio domain because these two tasks are all seq2seq tasks with variable length inputs.

**Summary Of The Paper:**

This paper proposes a new convolutional layer to achieve global convolution by kernel size upsampling. It's based on an existing study S4, and the authors relieve some of the design limitation of S4 and make it easier to use for different applications. Experiments on many sequence understanding tasks including the LRA data sets show the effectiveness of the proposed architecture.

**Summary Of The Review:**

The method in this paper have the following merits:
valuable extension over existing frameworks, easy to understand, easy to reproduce and could be of great impact, intensive experiments.

---

> ### Author Response · Authors · 2022-11-14
> **Response to Reviewer WDs3**
>
> We thank the reviewer for appreciating the idea of SGConv and for providing detailed comments. Here we would like to address the reviewer’s questions in order:
>
> 1. Details of SGConv: We updated the paper and elaborated on the details of SGConv. Please refer to the general response for detailed notes of the updates.
>
> 2. Audio experiments: We thank the reviewer for the suggestions, and we conducted experiments on the YouTubeMix dataset containing 4 hours of high-quality audio and 8 seconds chunks. We utilized the Sashimi [1] structure and replaced the S4 layer with the SGConv layer. Due to the time limitation and the restriction of computational resources, up to now, we have yet to finish the whole training process. Currently, at about 80% of the whole training steps, the SGConv-based Sashimi achieves 1.303 negative log-likelihood (NLL) scores on the test set, whereas S4 has a score of 1.294 with full training. For previous methods, SampleRNN [2] achieves 1.723, and WaveNet [3] achieves 1.449. SGConv-based Sashimi is ~1.2X times faster than S4-based Sashimi during the training phase. We will add this part to the paper after finishing the experiments.
>
>
>
> [1] Goel, Karan, Albert Gu, Chris Donahue, and Christopher Ré. "It's Raw! Audio Generation with State-Space Models." arXiv preprint arXiv:2202.09729 (2022).
>
> [2] Mehri, Soroush, Kundan Kumar, Ishaan Gulrajani, Rithesh Kumar, Shubham Jain, Jose Sotelo, Aaron Courville, and Yoshua Bengio. "SampleRNN: An unconditional end-to-end neural audio generation model." arXiv preprint arXiv:1612.07837 (2016).
>
> [3] Oord, Aaron van den, Sander Dieleman, Heiga Zen, Karen Simonyan, Oriol Vinyals, Alex Graves, Nal Kalchbrenner, Andrew Senior, and Koray Kavukcuoglu. "Wavenet: A generative model for raw audio." arXiv preprint arXiv:1609.03499 (2016).

---

### Official Review · Reviewer_oChP · 2022-11-01

**Confidence:** 4
**Correctness:** 4
**Technical Novelty And Significance:** 3
**Empirical Novelty And Significance:** 3
**Recommendation:** 6

**Clarity, Quality, Novelty And Reproducibility:**

*Clarity* : The writing is generally clear but, as pointed out before, seems sparse at places - some of the repetetive text could be ommitted in favor of more details about the model and the experiments.

*Quality and Novelty*: The proposed method is simple, novel and leads to modest improvments compared to strong and complex baselines.

*Reproducibility* : The experiments should be possible to reproducible without major issues.

**Strength And Weaknesses:**

**Strengths**:
1. A general-purpose CNN with strong performance on classification tasks that apparently provides benefits to attention-based models as well.
2. The method is simpler than previously proposed state of the art methods and provides modest empirical gains.
3. Emperical evaluation is comprehensive spanning sentence classification, speech recognition, image classification, language modeling.


**Weaknesses**: There are no major weakness that I can think of. Some minor comments are:
1. Authors could've provided more details regarding the model and the description seems sparse.
2. It'd be helpful to quickly discuss whether SGConv has a corresponding RNN view or not.
3. Seems like the authors need to make customized changes to the parameterization as described in Appendix A.1.


**Questions to the author**:
1. SGConv is not a state space (i.e. no RNN view) so its not immeditely clear from the text how to perform inference for autoregressive generation tasks such as Wikitext-103? Could you elaborate on how you do this in Table 2?

2. Table 2: From your description it seems that (after the kernel construction) the convolution computation is identical to S4? If yes, are these speedups due to faster kernel construction?

**Minor comments**:
- Pg 1: Pathfinder is not by Tay et al, its by Linsley et al.
- Fact 1: This might seem sudden and out of place to someone not familiar with S4.
- Equation 1: Please formally define Upsample, maybe in the Appendix. Upsampling with interpolation='linear' isn't clear enough.
- Pg 9: "..state transition matrix to be diagonal" : This was first done in "Diagonal State Spaces are Effective as Structured State Spaces" by Gupta, Gu and Berant.

**Summary Of The Paper:**

After the success of S4 on long range classification and generation tasks, there has been a series of works towards building equally performant but simpler models. In this work, authors propose a novel, simpler and faster parameterization of the convolutional kernels used by S4-like models. For a hyperparameter $d$, a 1D convolutional kernel $K$ of length $L$ is constructed as a concatenation of $N$ independent kernels $K=(K_0,..,K_{N-1})$ where each $K_i$ is a created by upsampling a parameter vector $w_i$ of original length $d$ to length $2^i*L/d$. Moreover each $K_i$ is scaled by a factor $\alpha^i$ to provide locality bias which the authors show is crucial for good performance. Authors compare their parameterization (SGConv) to S4 and show that this method not only leads to faster kernel construction but consistently outperforms S4 on several sequence classification tasks.

The authors then show that their SGConv layers can be used in tandem with Transformer layers (with local attention) matching strong baselines on causal LM and sentence-level classification tasks.

The authors also show that SGConv outperforms state of the art methods (ConnNeXT, ViT etc) on image classification tasks (Imagenet-1k) even after flattening the image as a 1D sequence.

**Summary Of The Review:**

Authors propose a general purpose model that is simpler than previous, more complex, baselines and demonstrate strong performance across multiple tasks and modalities. Certain parts of the paper require clarity and I am giving a lower score - I'll be happy to increase my score if the authors clarify the raised concerns.

---

> ### Author Response · Authors · 2022-11-14
> **Response to Reviewer oChP**
>
> We thank the reviewer for the detailed and helpful comments. The reviewer asked several insightful questions, and here we would like to address them in order:
>
> 1. Details and improvement of discussion: Following your suggestion, we have made several updates to improve the description of our methods. Please refer to the general response for detailed notes of the updates. Please let us know if there is anything unclear.
>
> 2. RNN views of SGConv: Thanks for pointing out this view of the S4 model. Since SGConv does not have the state-space-structured kernel as S4, the current version of SGConv cannot be used recursively as an RNN. Yet, SGConv can still naturally perform autoregressive generation (See the bullet below). We added the discussion to the part of incorporating SGConv into modern architectures in Section 3.2. Nevertheless, SGConv aims to provide an understanding of S4 / SSMs (and, more generally, CNNs) in a manner that encourages further exploration for the community. How to enable SGConv to act as RNN is an interesting future direction, and we believe more clever designs are needed, as in the “Transformers are RNNs: Fast Autoregressive Transformers with Linear Attention” paper.
>
> 3. Autoregressive tasks (e.g., WikiText-103): Using SGConv to perform autoregressive tasks follows a similar process to classic causal convolutional models (e.g., PixelCNN [3] and WaveNet [1]) and Transformers. Concretely, the convolution kernel is unidirectional, where the computation at the embedding of $i$-th is only computed based on tokens before $i$, and left zero padding is used for ignoring the overlength kernel. During generation, hidden states of past tokens are cached for fast calculation of the next token with a single convolution step. Please refer to Figures 2 and 3 of WaveNet for illustrations of how classic causal convolution works. We also added this discussion to the paper, and thanks for pointing out this important implementation detail.
>
> 4. Customized changes to the parameterization: As mentioned in Figure 3, as long as the decaying principle is satisfied, the performance only changes marginally. Similarly, the parameterizations in Appendix A.1 only bring up to 1% improvements for the two datasets, Image and PathX. Previous works also require customized hyperparameters for different tasks, as in Table 11 in the S4 paper [2].
>
> 5. Speedup of kernel construction: Yes, the speedups are for kernel constructions. After the kernel is constructed, both SGConv and S4 use FFT to perform the convolution.
>
> 6. Minor comments: We thanked the detailed comments for the paper and updated the paper accordingly.
>
> We thank the reviewer again. Your comments really helped us improve the paper. Please feel free to comment further if anything is unclear, and we sincerely hope you can raise the score if you find our modifications have answered your questions.
>
> Reference:
>
> [1] Oord, Aaron van den, Sander Dieleman, Heiga Zen, Karen Simonyan, Oriol Vinyals, Alex Graves, Nal Kalchbrenner, Andrew Senior, and Koray Kavukcuoglu. "Wavenet: A generative model for raw audio." arXiv preprint arXiv:1609.03499 (2016).
>
> [2] Gu, Albert, Karan Goel, and Christopher Ré. "Efficiently modeling long sequences with structured state spaces." arXiv preprint arXiv:2111.00396 (2021).
>
> [3] Van den Oord, Aaron, Nal Kalchbrenner, Lasse Espeholt, Oriol Vinyals, and Alex Graves. "Conditional image generation with PixelCNN decoders." Advances in neural information processing systems 29 (2016).

---

> > ### Comment · Reviewer_oChP · 2022-11-18
> > **Follow-up to author response 1**
> >
> > Thank you for making the recommended changes - the pseudo code adds a lot of clarity and also further highlights the simplicity of the method. Final remarks:
> > 1. *Decoding* : If thats the case,  might be good to mention the complexity during decoding.
> > 2. *Benchmarking vs S4*:  Table 2 caption says the running times are for entire blocks whereas section 4.1.3 says its for kernel computaion. Maybe clarify that times are for full blocks but the speedups are solely due to faster kernel constrruction. I am saying this because for large batches FFT part dominates.
> >
> > Taking all things into consideration and positioning this work with previous works I am maintianing my score of 6.

---

### Author Response · Authors · 2022-11-14
**General response and paper updates**

We sincerely thank all the reviewers and the area chairs for their efforts in reviewing our paper. Here, we summarize the major updates during the discussion period.

1. Improving the description of SGConv: Following the reviewers' suggestions, we refactored the description of SGConv in Section 3.2 and split it into 1) how to parameterize the SGConv kernel and 2) how to incorporate the SGConv module into modern architecture. We further added pointers to 1) illustrations of how to use SGConv in the Transformer and classic CNN models in Figure 6, 7; 2) a new illustration of how SGConv kernel is applied to the input of SGConv module with FFT in Figure 8; 3) a new Python style pseudo-code of the whole SGConv module in Appendix B.2. We also elaborated how to use SGConv for autoregressive tasks in the part of incorporating SGConv into modern architectures.

2. We reformulated Fact 1 with the background of how it relates to the S4 kernel following Reviewer oChP's helpful suggestion.

3. We added more related works, including other ICLR submissions concurrent to our paper, for better positioning our work.

---

### Decision · Program_Chairs · 2023-01-20

**Decision:**

Accept: poster

**Justification For Why Not Higher Score:**

My recommendation leans heavily on the evaluation from reviewer oChP, who I personally recruited as a reviewer and who is an expert on state-space models. The fact that oChP was not willing to raise their score after the rebuttal and discussion says to me that this paper is best presented as a poster. If the SACs and PCs feel that this paper is one of the stronger follow-on works to S4 at ICLR, I would not be opposed to making the paper a spotlight or oral.


**Justification For Why Not Lower Score:**

This paper brings useful insights to state-space models and the claims of the paper are well supported by empirical results. The community will benefit from publication of this work.

**Metareview: Summary, Strengths And Weaknesses:**

# Summary
Inspired by the interpretation of the S4 model as a global convolutional model (a model in which the convolutional kernel is of the same size as the input), this paper attempts to distill usable principles for the design of the kernel from S4 and then test those principles on the long range arena (LRA) and Speech Commands tasks. The paper proposes two design principles, (1) parameterization that grows more slowly than sequence length and (2) a locality bias implemented with decaying kernels, and realizes them with a Structured Global Convolution (SGConv) model. Tests on LRA and Speech Commands show that the SGConv model outperforms S4 in most cases. Speed tests show that inference with SGConv is faster than with S4, thanks to the faster kernel construction of SGConv. Additional tests on image recognition, sentence classification, and language modeling illustrate the potential for SGConv to be used as a drop-in replacement for other components when the modeling of long-range dependencies is important.

# Strengths
- Nice analysis of the important principles for effective long-range modeling using global convolutional models.
- Empirical results are strong, and the evaluation of the proposed model is strong.

# Weaknesses
- The paper should be a bit clearer on the point that the inference speed advantage enjoyed by SGConv over S4 is due to the faster kernel construction process.


**Note From Pc:**

if the above contains the word "oral" or "spotlight" please see: "oral" presentation means -> notable-top-5% and "spotlight" means -> notable-top-25%. As stated in our emails, we are disassociating presentation type from AC recommendations

**Summary Of Ac-Reviewer Meeting:**

N/A